# Timeliness of routine childhood vaccination in low- and middle-income countries, 1978–2021: Protocol for a scoping review to map methodologic gaps and determinants

Oghenebrume Wariri [1,2]*, Uduak Okomo[1], Yakubu Kevin Kwarshak[3], Kris A. Murray[4,5], Chris Grundy[2‡], Beate Kampmann[1,6‡]

1 Vaccines and Immunity Theme, MRC Unit The Gambia at the London School of Hygiene and Tropical Medicine, Fajara, The Gambia, 2 Department of Infectious Disease Epidemiology, London School of Hygiene and Tropical Medicine, London, United Kingdom, 3 Department of Surgery, Jos University Teaching Hospital, Jos, Nigeria, 4 MRC Unit The Gambia at the London School of Hygiene and Tropical Medicine, Fajara, The Gambia, 5 MRC Centre for Global Infectious Disease Analysis, Imperial College School of Public Health, Imperial College London, London, United Kingdom, 6 The Vaccine Centre, London School of Hygiene and Tropical Medicine, London, United Kingdom

‡ These authors are joint senior authors on this work.
* Oghenebrume.Wariri@lshtm.ac.uk

**Funding:** This project is part of the EDCTP2 programme supported by the European Union (grant number TMA2019CDF-2734 - TIMELY). OW is also supported by an Imperial College London Wellcome Trust Institutional Strategic Support Fund (ISSF) (grant no RSRO_P67869). KM is

## Abstract

The literature on the timeliness of childhood vaccination (i.e. vaccination at the earliest appropriate age) in low-and middle-income countries has important measurement and methodological issues that may limit their usefulness and cross comparison. We aim to conduct a comprehensive scoping review to map the existing literature with a key focus on how the literature on vaccination timeliness has evolved, how it has been defined or measured, and what determinants have been explored in the period spanning the last four decades. This scoping review protocol was developed based on the guidance for scoping reviews from the Joanna Briggs Institute. We will include English and French language peer-reviewed publications and grey literature on the timeliness of routine childhood vaccination in low-and middle-income countries published between January 1978 through to 2021. A three-step search strategy that involves an initial search of two databases to refine the keywords, a full search of all included electronic databases, and screening of references of previous studies for relevant articles missing from our full search will be employed. The search will be conducted in five electronic databases: MEDLINE, EMBASE, Global Health, CINAHL and Web of Science. Google search will also be conducted to identify relevant grey literature on vaccination timeliness. All retrieved titles from the search will be imported into Endnote X9.3.3 (Clarivate Analytics) and deduplicated. Two reviewers will screen the titles, abstracts and full texts of publications for eligibility using Rayyan–the web based application for screening articles for systematic reviews. Using a tailored data extraction template, we will extract relevant information from eligible studies. The study team will analyse the extracted data using descriptive statistical methods and thematic analysis. The results will be presented using tables, while charts and maps will be used to aid the visualisation of the key findings and themes. The proposed review will generate evidence on key methodological

supported by joint Centre funding from the UK Medical Research Council and Department for International Development [MR/R0156600/1]. The Vaccines and Immunity Theme (OW, UO, and BK) is jointly funded by the UK MRC and the UK Department for International Development (DFID) under the MRC/DFID Concordat agreement and is also part of the EDCTP2 Programme supported by the EU (MC UP_A900/1122, MC UP A900/115).

**Competing interests:** The authors have declared that no competing interests exist.

gaps in the literature on timeliness of childhood vaccination. Such evidence would shape the direction of future research, and assist immunisation programme managers and country-level stakeholders to address the needs of their national immunisation system.

## Introduction

Since the World Health Organization (WHO) introduced the Expanded Programme on Immunization (EPI) in 1974 [1], the proportion of children protected against vaccine-preventable diseases (VPDs) continue to increase with more than a billion children vaccinated in the last decade alone [2]. Globally, about 2–3 million deaths from diseases such as diphtheria, tetanus, pertussis and measles are prevented yearly with lifesaving childhood vaccines [2]. In low- and middle-income countries (LMICs), current estimates suggest that between 2000 and 2019, 36 million deaths have been averted among children under 5 by vaccination programmes [3]. Although EPI has drastically reduced the incidence of, and deaths from VPDs, its success across and within countries vary, especially in LMICs.

The usual metric employed for assessing the success of immunisation systems is routine vaccination coverage at specific ages [4]. This metric, however, does not take into consideration whether the vaccines have been received in a timely manner, in accordance with the recommended national vaccination windows. Even in the presence of high overall coverage rates, measurement of crude vaccination coverage can mask substantial delays in vaccinations [5]. Timeliness of vaccination (i.e. vaccination at the earliest appropriate age) matters because vaccinations that are received too early or too closely spaced may result in suboptimal immunological responses [6]. On the other hand, delayed childhood vaccination unnecessarily prolongs exposures to VPDs such as pertussis, measles and *Haemophilus influenzae* type b– diseases for which peaks and severity are worse during infancy [6, 7]. Untimely vaccination, therefore, endangers the health of children and compromises herd immunity, with potential implications for VPDs outbreaks irrespective of coverage rates.

Although there is a growing body of literature on timeliness of childhood vaccinations, many studies have focussed on high-income countries where VPD burden is comparatively low. Furthermore, the literature from LMICs have important measurement and methodological issues which may limit their usefulness and cross comparison. For example, there is a lack of a measurement cut-off or agreed-upon definition for what might be considered timely vaccination [8]. While some authors have studied vaccination timeliness using a continuous measure [9–11], others have used categorical, but with varying cut-offs points [12–14]. Second, the determinants of vaccination timeliness have not been robustly researched in the empirical literature which makes it difficult to more clearly define the priority for future research and policy.

To our knowledge, the systematic review by Masters et. al. (2019) was the first to summarise the literature on vaccination timeliness in LMICs to identify methodological gaps and provided recommendations for future studies [8]. While their review has provided important insights into the lack of a uniform definition of what might constitute timely vaccination, there were several limitations that have necessitated a further review. First, EPI was introduced by WHO in 1974 and by 1977 all LMICs had been mandated to adopt the WHO-recommended schedule [1]. The global COVID pandemic has been shown to negatively affect EPI vaccine delivery and acceptance, especially in LMICs. By limiting their review to studies conducted between 2007 and 2017 therefore, important studies conducted before 2007 and after

2017 would have been omitted. Second, their review was conducted in only three electronic databases and restricted to studies published in English language. To bridge this gap, we therefore aim to conduct a more comprehensive scoping review, and map the existing literature on vaccination timeliness with a key focus on the methodological gaps in its definition, measurement, and determinants.

## Methods

This protocol was developed based on the guidance for scoping reviews from the Joanna Briggs Institute (JBI) [15]. The scoping review process will be guided by the methodological framework proposed by Arksey and O'Malley [16]. The reporting of the scoping review output will be conducted using the Preferred Reporting Items for Systematic Reviews and Meta-Analyses extension for Scoping Reviews (PRISMA-ScR) checklist [17].

### Review questions

This scoping review will answer the following key research questions:

1. How has the literature on childhood vaccination timeliness in LMICs evolved (i.e. studies published per year and the antigens studied over time) in the last four decades?

2. In what LMIC countries have the literature on childhood vaccination timeliness been focused?

3. How has childhood vaccination timeliness been defined or measured in the empiric studies from LMICs in the last four decades?

4. What statistical analytic approaches have been used in the literature to assess childhood vaccination timeliness?

5. What determinants or factors contributing to untimely childhood vaccination have been studied in LMICs?

### Information sources

We plan this review to identify peer-reviewed and online grey literature on vaccination timeliness in any low-and middle-income country (LMIC) [18]. The search will be conducted in five electronic databases: MEDLINE, EMBASE, Global Health, CINAHL and Web of Science. Using selected terms from the search strategy, Google search will also be conducted to identify relevant grey literature on vaccination timeliness.

### Search strategy

As recommended by the JBI, a three-step search strategy will be utilised to ensure that our search is comprehensive [15]. The search strategy was developed in consultation with, and refined based on input from a librarian. First, a preliminary search of MEDLINE and Web of Science was conducted on March 27, 2021 using the key concepts: *Childhood; Vaccination*; *Timeliness*; and *LMICs*. To further refine the search strategy, the initial search was followed by an analysis of the text words in the title and abstract of the retrieved papers and the index terms used in describing the articles. An example of the search strategy and terms used in MEDLINE is included as S1 Table in this protocol. The second step will be a search conducted across all five included databases using the search strategy which has been refined based on all identified keywords and index terms from the first step. The search strategy will be adapted

**Step 3**
Search of *reference list* of retrieved articles and reports for relevant additional sources

**Step 2**
Apply refined search strategy to *MEDLINE, EMBASE, Global Health, CINAHL and Web of Science*

**Step 1**
Preliminary search of *MEDLINE* and *Web of Science* to further refine the search strategy

**Fig 1. The three-step search strategy that will be utilised to ensure a comprehensive search for the scoping review.**

based on the search terminology for each of the included databases. In the third step, the reference list of all the identified papers and reports will be searched for additional sources. See Fig 1 for illustration of the search strategy.

## Inclusion criteria

To ensure comprehensiveness, quantitative or mixed-methods studies or reports will be included if they meet the following criteria: (a) focused on childhood vaccinations that are part of the routine national EPI programme; (b) calculate some measure of timeliness related to vaccine coverage; (c) are conducted on data from countries categorised as LMICs by the World Bank [18]; (d) published in English or French languages; and (e) from January 1978 through to 2021. The decision to restrict this scoping review to studies conducted in LMICs is because of the higher burden of VPDs in these countries and the fact that the national EPI schedule in these countries adopts the WHO-recommended routine childhood immunization schedule, in contrast to many high-income countries. The choice to include studies published from January 1978 is based on the fact that routine childhood immunization against diphtheria, pertussis, tetanus, poliomyelitis, measles and tuberculosis in LMICs commenced in 1977 in many countries [1]. The search will be extended to 2021 to ensure that the latest evidence on vaccination timeliness is included in this review even as the ongoing COVID-19 pandemic has impacted on routine vaccination programmes with potential delayed vaccinations in many LMICs.

## Exclusion criteria

Systematic reviews, study protocols, correspondences, journal commentaries, and conference abstracts will be excluded. Additionally, studies which are based on the modelling of vaccination timeliness will also be excluded.

## Study selection

All retrieved titles from the search will be imported into Endnote X9.3.3 (Clarivate Analytics) and de-duplication of records will be performed using the Endnote duplicates function. The references will then be exported to Rayyan (a web based application for screening articles for systematic reviews) where two reviewers will screen the titles and abstracts for relevance [19]. In this initial stage, two reviewers will independently screen the titles and abstracts to identify which studies meet eligibility criteria after which the included references will be exported back to Endnote for full-text screening and extraction. In the second stage, one out of the first two reviewers that performed the initial assessment will screen the full-text of the included studies to verify if they will be appropriate for full data extraction while the second reviewer will verify all decisions. During this stage, some articles will be excluded from full data extraction if they do not meet the inclusion/exclusion criteria. The pre-specified inclusion criteria in this proto-col will guide article selection for inclusion. All decisions related to article inclusion will be made through consensus by the two reviewers conducting the extraction. However, if the two reviewers fail to reach a consensus, a third member of the review team will be consulted to help resolve the disagreement. The process and outcome of screening, inclusion, and exclusion of articles will be illustrated using the PRISMA flow chart diagram for reporting items for sys-tematic reviews.

## Data extraction

A data extraction template has been developed which will be used to record the information of interest from the included articles. This template was adapted from the JBI data extraction tool for scoping reviews [20]. Two members of the review team have piloted and refined the data extraction template on 20 randomly selected articles during the protocol development stage as recommended by Arksey & O'Malley [16] and the JBI [20]. The key information to be extracted is listed in Box 1 below. During the full data extraction process, one reviewer will extract the data while another reviewer will verify the extracted data to ensure the quality of the data. Critical appraisal of the included studies will not be conducted because it is not man-datory for scoping reviews [20].

### Box 1. Key information in the data extraction template

1. Author (lead author only and et.al.)
2. Year of study publication
3. Source/country of origin of the study (list all the countries)
4. Study population (i.e. age range of children included)
5. Methodology or study design (e.g. cross sectional, cohort, etc.)
6. Dataset used (e.g. Health survey data, surveillance data, etc.)
7. Routine EPI vaccines/antigens studied (i.e. indicate names of the antigen)
8. How vaccination timeliness was measured (e.g. continuous measures, categorical measures, etc.)

9. Statistical analysis approach employed

10. Determinants or factors contributing to vaccination timeliness that were explored

## Presentation and charting of results

The extracted data will be analysed using descriptive statistical methods. The results will be presented using tables, while charts and maps will be used to aid the visualisation of the key findings. The information to be captured with a table include the lead author, study population, study design, the dataset used among other variables. The year of study publication will be summarised using a line graph showing trends since 1978, while the number of studies published per country will be represented using a thematic map. The determinants of vaccination timeliness will be organised according to *a priori* categories that have been developed based on the three-delays conceptual framework by Thaddeus and Maine [21]. All results will be presented using a narrative summary according to the objectives of this scoping review.

## Ethics

Ethical approval is not required for scoping review because it involves the synthesis of publicly available publications. Pre-registration in a public registry such as PROSPERO is not mandatory for scoping review protocols.

## Discussion

The proposed scoping review is expected to map the existing literature on the timeliness of vaccination in LMICs from 1978 through 2021, with a focus on how the literature has evolved, in what geographic context, its definition, and determinants. Specifically, the review seeks to map how timeliness of childhood vaccination has been conceptualised or measured in the literature. Mapping the evidence on how vaccination timeliness has been measured in LMICs over the past four decades will highlight critical methodological gaps that will aid future research to adopt a more robust measurement of vaccination timeliness.

Mapping the evidence to show which determinants have been previously or more routinely explored in the literature will highlight the potential research gaps related to the determinants of childhood vaccination timeliness. There is emerging evidence that shows that supply-side factors such as geographic accessibility (travel time, distance to facility, etc.) to immunisation service points impacts the likelihood of receiving childhood vaccination. Yet, to the best of our knowledge, the influence of geographic accessibility on the timeliness of childhood vaccination has been less explored in the literature in LMIC [22, 23]. Such a gap limits the availability of critical evidence that could assist immunisation programme managers and country-level stakeholders to address the needs of EPI.

A limitation of this scoping review is that it will not include studies from high-income countries, and studies that are based on vaccinations not given within the remit of the routine EPI schedule such as those given in adolescence, adulthood, and even the recent COVID-19 vaccination. While it is important to study the timeliness of vaccination in these contexts, we will focus on routine childhood vaccination in LMICs for two reasons. First, LMICs have the highest burden of VPDs which makes it imperative for the EPI vaccines to be received within the predetermined vaccination windows. Second, the peak and severity of VPDs is worse

during early childhood or infancy which further highlight the need for receipt of vaccines against VPDs in an age-appropriate manner, before the peak of exposures. Despite the limitations highlighted above, the proposed scoping review, when completed, will provide robust evidence on the methodological gaps in the literature on vaccination timeliness in LMICs spanning more than four decades. The results would aid the design and conduct of future empirical studies into the timeliness of routine childhood vaccinations, thus, ensuring the usefulness and cross comparison of their output.

## Supporting information

**S1 Table. Example of the full search strategy and terms developed for use in MEDLINE.** (DOCX)

**S1 Checklist. PRISMA-ScR fillable checklist.** (DOCX)

## Author Contributions

**Conceptualization:** Oghenebrume Wariri.

**Funding acquisition:** Oghenebrume Wariri.

**Investigation:** Oghenebrume Wariri, Uduak Okomo, Yakubu Kevin Kwarshak.

**Methodology:** Oghenebrume Wariri, Uduak Okomo, Yakubu Kevin Kwarshak, Kris A. Murray, Chris Grundy, Beate Kampmann.

**Project administration:** Oghenebrume Wariri.

**Resources:** Oghenebrume Wariri.

**Supervision:** Uduak Okomo, Kris A. Murray, Chris Grundy, Beate Kampmann.

**Validation:** Yakubu Kevin Kwarshak.

**Writing – original draft:** Oghenebrume Wariri.

**Writing – review & editing:** Oghenebrume Wariri, Uduak Okomo, Yakubu Kevin Kwarshak, Kris A. Murray, Chris Grundy, Beate Kampmann.

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
