## [Decision Letter · Decision Letter 0]

28 May 2021

PONE-D-21-12294

Timeliness of childhood vaccination in low-and middle-income countries, 1978 - 2021: protocol for a scoping review to map methodologic gaps and determinants

PLOS ONE

Dear Dr. Wariri,

Thank you for submitting your manuscript to PLOS ONE. After careful consideration, we feel that it has merit but does not fully meet PLOS ONE’s publication criteria as it currently stands. Therefore, we invite you to submit a revised version of the manuscript that addresses the points raised during the review process.

We look forward to receiving your revised manuscript.

Kind regards,

Charles Shey Wiysonge, MD, PhD

Academic Editor

PLOS ONE

Journal Requirements:

Additional Editor Comments:

None

Reviewers' comments:

Reviewer's Responses to Questions

**Comments to the Author**

1. Does the manuscript provide a valid rationale for the proposed study, with clearly identified and justified research questions?

Reviewer #1: Yes

2. Is the protocol technically sound and planned in a manner that will lead to a meaningful outcome and allow testing the stated hypotheses?

Reviewer #1: Yes

3. Is the methodology feasible and described in sufficient detail to allow the work to be replicable?

Reviewer #1: Yes

4. Have the authors described where all data underlying the findings will be made available when the study is complete?

Reviewer #1: Yes

5. Is the manuscript presented in an intelligible fashion and written in standard English?

Reviewer #1: Yes

6. Review Comments to the Author

You may also provide optional suggestions and comments to authors that they might find helpful in planning their study.

Reviewer #1: Apart from the one comment and suggested addition, the paper is sound, cohesive, relevant and well presented.

7. PLOS authors have the option to publish the peer review history of their article (what does this mean?). If published, this will include your full peer review and any attached files.

Reviewer #1: **Yes: **Joshua Karras

---

## [Author Response · Author response to Decision Letter 0]

30 May 2021

Journal requirements

1. Please ensure that your manuscript meets PLOS ONE's style requirements, including those for file naming. The PLOS ONE style templates can be found HERE and HERE

Response: Thanks for providing the guidelines. We have revised the manuscript to meet the PLOS ONE’s style requirement. All edits are in track changes in the document marked ‘’Revised Manuscript with Track Changes’.

2. Please review your reference list to ensure that it is complete and correct. If you have cited papers that have been retracted, please include the rationale for doing so in the manuscript text, or remove these references and replace them with relevant current references. Any changes to the reference list should be mentioned in the rebuttal letter that accompanies your revised manuscript. If you need to cite a retracted article, indicate the article’s retracted status in the References list and also include a citation and full reference for the retraction notice

Response: We have reviewed the reference list as requested and can confirm that it is complete and correct. Specifically, we did not make any changes to the reference list and we have not cite any papers which have been retracted.

Comments for the Authors (Reviewer #1)

1. Apart from the one comment and suggested addition, the paper is sound, cohesive, relevant and well presented.

Response: We thank the reviewer for the time spent in reviewing our manuscript and for providing very positive comments about our scoping review protocol.

We have provided a point-by-point response to the reviewers’ comment below.

2. To increase usefulness of the paper and more comprehensive opportunities for future studies, a 6th review question may be included:

 What determinants or factors contributing to untimely childhood vaccination have identified but not studied in LMICs?

Response: We appreciate the reviewer for this thoughtful comment and for recommending the addition of an additional question to the scoping review protocol. While we agree with the reviewer in principle, we did not include this specific question for the reasons we have explained below.

Any determinants or factors contributing to untimely routine childhood vaccinations that have been identified but was not studied in LMICs can only be identified after the completion of this scoping review. We cannot at this point identify such determinants without first conducting a scoping review of the literature on timeliness of vaccinations across LMIC and HIC. Additionally, the included papers and manuscript will not primarily have that information (i.e., each paper will not tell us ‘which determinants already identified that was not studied’). Thus, this question can only be answered deductively after completing the full review. We aim to thoroughly contextualise such factors in the discussion section of the full manuscript by comparing our findings (i.e., the determinants of timeliness identified from our scoping review) to those from studies in HICs.

---

## [Editor Report · Decision Letter 1]

7 Jun 2021

Timeliness of routine childhood vaccination in low-and middle-income countries, 1978 - 2021: protocol for a scoping review to map methodologic gaps and determinants

PONE-D-21-12294R1

Dear Dr. Wariri,

We’re pleased to inform you that your manuscript has been judged scientifically suitable for publication and will be formally accepted for publication once it meets all outstanding technical requirements.

Kind regards,

Charles Shey Wiysonge, MD, PhD

Academic Editor

PLOS ONE

---

## [Editor Report · Acceptance letter]

9 Jun 2021

PONE-D-21-12294R1 

Timeliness of routine childhood vaccination in low-and middle-income countries, 1978 - 2021: protocol for a scoping review to map methodologic gaps and determinants 

Dear Dr. Wariri:

I'm pleased to inform you that your manuscript has been deemed suitable for publication in PLOS ONE. Congratulations! Your manuscript is now with our production department. 

Kind regards, 

on behalf of

Professor Charles Shey Wiysonge 

Academic Editor

PLOS ONE